# The Effect of the Addition of *Apulian black* Chickpea Flour on the Nutritional and Qualitative Properties of Durum Wheat-Based Bakery Products

**DOI:** 10.3390/foods8100504

**Published:** 2019-10-16

**Authors:** Antonella Pasqualone, Davide De Angelis, Giacomo Squeo, Graziana Difonzo, Francesco Caponio, Carmine Summo

**Affiliations:** Department of Soil, Plant and Food Science (DISSPA), University of Bari Aldo Moro, Via Amendola, 165/a, I-70126 Bari, Italy; davide.deangelis@uniba.it (D.D.A.); giacomo.squeo@uniba.it (G.S.); graziana.difonzo@uniba.it (G.D.); francesco.caponio@uniba.it (F.C.); carmine.summo@uniba.it (C.S.)

**Keywords:** pulses, re-milled semolina, bread, pizza, focaccia, rheological properties, reofermentograph, bioactive compounds, texture, sensory profile

## Abstract

Historically cultivated in Apulia (Southern Italy), *Apulian black* chickpeas are rich in bioactive compounds such as anthocyanins. This type of chickpea is being replaced by modern cultivars and is at risk of genetic erosion; therefore, it is important to explore its potential for new food applications. The aim of this work was to assess the effect of the addition of *Apulian black* chickpea wholemeal flour on the nutritional and qualitative properties of durum wheat-based bakery products; namely bread, “focaccia” (an Italian traditional bakery product similar to pizza), and pizza crust. Composite meals were prepared by mixing *Apulian black* chickpea wholemeal flour with re-milled semolina at 10:90, 20:80, 30:70, and 40:60. The rheological properties, evaluated by farinograph, alveograph, and rheofermentograph, showed a progressive worsening of the bread-making attitude when increasing amounts of chickpea flour were added. The end-products expanded less during baking, and were harder and darker than the corresponding conventional products, as assessed both instrumentally and by sensory analysis. However, these negative features were balanced by higher contents of fibre, proteins, and bioactive compounds, as well as higher antioxidant activity.

## 1. Introduction

The chickpea (*Cicer arietinum* L.) plays an important role in a healthy and environmentally sustainable diet. In fact, as with other pulses, the chickpea fixes atmospheric nitrogen, thus increasing soil fertility [1]. This species is rich in proteins, and particularly if consumed in combination with cereals to compensate for amino acid deficiencies, can help decrease the dietary intake of meat.

Chickpea is commonly classified in two main types: *kabuli*, characterized by large seeds with beige coats, and *desi*, characterized by small and rough seeds with a black or brown coat [2]. However, there is another type of chickpea, historically cultivated in Apulia (South of Italy), apparently similar to *desi* because of its black coat, but different from the genetic point of view [3]. This *Apulian black* type, which is being replaced by modern cultivars, and is therefore, at risk of genetic erosion [3], has an interesting potential for further commercial development due to its high content of antioxidant compounds, such as anthocyanins and carotenoids [4,5].

Both traditional and new food uses of pulses have been proposed in recent years [6]. Chickpea flour has been used as an ingredient in the production of vegetable beverages [7], extruded snacks [8], canned purée [9], pasta [10], and gluten-free bread [11]. Several attempts have been made to also use chickpea flour in conventional bread-making, rediscovering an ancient tradition of Albania and Turkey, where chickpea bread was commonly prepared in the past [12]. The addition of 10–30% chickpea wholemeal flour to common wheat flour was proposed, but a negative effect on bread quality in terms of volume, internal structure, texture, and sensory acceptability was observed [13,14,15]. Blends of different pulses (chickpeas, lentils, and beans, with chickpeas accounting for only 5% of the total flour) was then proposed [16] for sourdough, the latter being helpful to reduce antinutrients such as phytates. The use of dried chickpea sourdough in bread-making was studied, pointing out that 10% was the optimal amount [17]. Together with the addition of 5–30% fermented chickpea flour, the use of 0.5–3% xanthan gum was found to improve the viscoelastic properties of dough [18]. Alternatively, some improvers, such as ascorbic acid and sodium stearoyl-2-lactylate, were helpful when using 25–35% chickpea flour in combination with wheat wholemeal flour [19]. In this frame, however, only one study proposed to enrich durum wheat re-milled semolina with flour of pulses, namely yellow pea, for preparing bread [20], and no research was made to employ chickpea flour in durum wheat bread-making. Moreover, no study considered the use of pigmented chickpeas.

Durum wheat (*Triticum turgidum* var. *durum*) is largely used in the production of pasta and cous cous, and is characterized by tenacious gluten and by the presence of carotenoid pigments. The latter are important from the nutritional point of view due to provitamin A activity, and confer the typical yellowish colour to the end-products, much appreciated by consumers. In the Mediterranean area, part of durum wheat production is used to prepare traditional bread and other bakery products, with a typically dense crumb [21]. In particular, durum wheat re-milled semolina has to be used, which has a particle size similar to that of bread wheat flour (i.e., smaller than that of semolina used for pasta-making) [22], ensuring high hydration rate.

Pizza, originated in Italy, has become a popular food worldwide. The demand of pizza has continued to grow in recent decades, so that food industrial companies have shown growing interest in its production [23]. Pizza is usually prepared with bread wheat flour, but other types of flour have been proposed in the recent years, including durum wheat re-milled semolina [24]. Pizza crust is a convenient food that can be seasoned at home.

“Focaccia” is another Italian traditional bakery product, widely consumed as street food, similar to pizza but containing higher amounts of oil [25]. It may be defined as a leavened greasy flat bread varyingly seasoned, the most typical topping being cherry tomatoes and olives, accompanied by the rosemary and the potato and onion variants. Additionally, in the preparation of focaccia, the use of durum wheat re-milled semolina is quite common [25].

To the best of our knowledge, no papers have considered the enrichment of durum wheat bread, pizza, and focaccia with black chickpea flour. The aim of this work was, therefore, to assess the effect of the addition of *Apulian black* chickpea flour on the nutritional and qualitative properties of durum wheat based bakery products; namely, bread, focaccia, and pizza crust. In particular, a wholemeal flour of chickpeas was used, for accomplishing the current dietary guidelines that highlight the need of increasing fibre intake.

## 2. Materials and Methods

### 2.1. Materials

*Apulian black* chickpeas (*C. arietinum* L.) were supplied by CerealPuglia s.r.l. (Altamura, Italy). Durum wheat (*T. turgidum* var. *durum*) re-milled semolina was supplied by the milling company Industria Molitoria Mininni s.r.l. (Altamura, Italy). Extra virgin olive oil was supplied by Agridè (Bitonto, Italy). Kastalia stabilized liquid yeast, composed of *Saccharomyces cerevisiae*, salt, and xanthan gum as its stabilizer, and having fermentative power >120 mL CO_2_ at 20 °C and 1 atm, was provided by Lesaffre Italia (Trecasali, Parma, Italy).

### 2.2. Preparation of the Composite Flours

*Apulian black* chickpeas were ground at the Food Science laboratory of the University of Bari using a laboratory-scale mill (ETA, Vercella Giuseppe, Mercenasco, Italy) equipped with a sieve of 0.6 mm, to obtain wholemeal flour. Durum wheat re-milled semolina was then used to prepare composite meals containing 10, 20, 30, and 40/100 g of *Apulian black* chickpea wholemeal flour.

### 2.3. Formulation of the Bakery Products

Three types of bakery products, namely bread, focaccia, and pizza crust were produced using a composite meal made of re-milled semolina (60/100 g) and *Apulian black* chickpea wholemeal flour (40/100 g). For each product, a control made of pure re-milled semolina was prepared. The formulation, reported in Table 1, was the same for the three types of products, except for the amount of oil, which was not used for preparing bread according to the traditional formulation of Italian durum wheat bread [21]. The quantity of water was added to flour in quantities sufficient to reach a dough consistency of 500 BU, assessed by preliminary farinograph analyses (Brabender instrument, Duisburg, Germany). The preparation of the experimental bakery products was carried out at the bakery laboratory “Buéne” of the Industria Molitoria Mininni s.r.l. (Altamura, Italy).

### 2.4. Preparation of Bread

Flour, water, and yeast were kneaded for 6 min by a spiral mixer (Mecnosud, Flumeri, Italy). Then, salt was added and kneading was continued for other 6 min. The formulation was as in Table 1. The homogeneous dough obtained was left to rise for 1.5 h at 35 °C, RH = 20% (Electric oven-leavening combo EKL 1264 TCR, Tecnoeka S.r.l., Borgoricco, Italy), then was divided into 110 g portions manually shaped at 14 cm length, 6 cm width, and 0.5 cm thickness (“ciabatta” bread type), and left to rise again for 30 min at 35 °C, RH = 20% (EKL 1264 proofer, Tecnoeka S.r.l., Borgoricco, Italy). Bread was finally baked at 220 °C for 20 min in an electric oven (Smeg SI 850 RA-5 oven, Smeg S.p.A., Guastalla, Italy).

### 2.5. Preparation of Focaccia

Flour, water, and yeast were kneaded for 6 min by a spiral mixer (Mecnosud, Flumeri, Italy). Then, salt and oil were added and kneading was continued for other 6 min. The formulation was as in Table 1. The homogeneous dough obtained was left to rise for 1 h and 15 min at 35 °C, RH = 20% (EKL 1264 proofer, Tecnoeka S.r.l., Borgoricco, Italy), then was divided into 110 g portions of spherical shape, which were manually flattened at 1.5 cm thickness and about 13 cm diameter, put in disposable aluminum pans, previously oiled with other 10 g of extra virgin olive oil, and left to rise again for 1 h and 15 min at 35 °C, RH = 20% (EKL 1264 proofer, Tecnoeka S.r.l., Borgoricco, Italy). Focaccia was finally baked at 220 °C for 15 min in an electric oven (Smeg SI 850 RA-5 oven, Smeg S.p.A., Guastalla, Italy).

### 2.6. Preparation of Pizza Crust

Flour, water, and yeast were kneaded for 6 min by a spiral mixer (Mecnosud, Flumeri, Italy). Then, salt and oil were added and kneading was continued for other 6 min. The formulation was as in Table 1. The homogeneous dough obtained was divided into 220 g portions of spherical shape which were left to rise for 1 h and 45 min at 35 °C, RH = 20% (EKL 1264 proofer, Tecnoeka S.r.l., Borgoricco, Italy). Subsequently, the dough portions were manually flattened to the thickness of 0.5 cm and diameter of about 28 cm and were immediately baked at 380 °C for 3 min in an electric oven for pizza, equipped with a refractory cooking stone (G3 pizza oven, Ferrari, Rimini, Italy).

### 2.7. Chemical Analyses

Protein (N × 5.7), ash, and moisture contents were determined according to the American Association of Cereal Chemists (AACC) methods 46-11.02, 44–19, and 08–01, respectively [26]. Fat was extracted and determined by Soxhlet apparatus using diethyl ether as solvent. Total dietary fibre was determined by the enzymatic-gravimetric procedure [27]. Carbohydrates were calculated by difference: 100 − (moisture + proteins + lipids + fibre + ash). Energy value (kJ) was calculated by Atwater general conversion factors, by considering the contribution of 8 kJ/g from total dietary fibre also, in accordance with the Annex XIV of the Regulation (EC) number 1169/2011 [28]. Total anthocyanins, total phenolic compounds and antioxidant activity by DPPH method were determined as in [29]. The antioxidant activity by ABTS method was assessed as in [30]. Total carotenoid pigments were determined according to AACC method 14–50.01 [26]. All analyses were carried out in triplicate.

### 2.8. Determination of the Rheological Properties and Fermentative Attitude of Flours and Composite Meals

The farinograph indices were determined according to the AACC 54–21 method [26] by a farinograph (Brabender instrument, Duisburg, Germany), equipped with the software Farinograph (Brabender instrument, Duisburg, Germany). Alveograph trials were performed according to the AACC method 54–30A [24] using an alveoconsistograph, equipped with the software Alveolink NG (Tripette et Renaud, Villeneuve-la-Garenne, France). The α-amylase activity was determined by using the Falling Number 1500 apparatus (Perten Instruments AB, Huddinge, Sweden), according to the ISO 3093:2009 method [31]. Rheofermentometer analysis (F3 rheofermentometer, Tripette et Renaud, Chopin Technologies, Villeneuve-la-Garenne, France) was carried out according to the AACC 89-01 method [26] at 28.5 °C for 3 h, with a 2000 g weight. All analyses were carried out in triplicate.

### 2.9. Physical Determinations of Bakery Products

Texture profile analysis (TPA) was performed on bread and focaccia. Pizza crust was not analyzed due to its very low thickness (0.7–1.0 cm). A Z1.0 TN texture analyzer (Zwick Roell, Ulm, Germany) was used, equipped with a stainless steel square probe (4 cm side) and a 50 N load cell. Data were acquired by means of the TestXPertII version 3.41 software (Zwick Roell, Ulm, Germany). Two centimeter thick slices (3.5 cm × 3.5 cm) were prepared and analyzed. The TPA conditions in the cyclic compression test were: 1 mm/s probe compression rate; 40% sample deformation in both the compressions; and a 5 s pause before second compression. The analyses were carried out in triplicate.

The color indices *L**, *a**, and *b** were measured by using a Chromameter CM-600d (Konica Minolta, Tokyo, Japan). The brown index was calculated as 100 − *L**. Five replicated analyses were made.

The respective diameter (D), length (L), width (W), and thickness (T) values of bread, focaccia, and pizza crust before and after baking were determined by a caliper and used to calculate the percentage variation due to baking as follows:

% of variation of D (or L, W, T) = [D (or L, W, T) after baking − D (or L, W, T) before baking]/D (or L, W, T) before baking × 100. The analyses were carried out in triplicate.

### 2.10. Sensory Analysis

Quantitative descriptive analysis (QDA) of bread, focaccia, and pizza crust respectively, was performed by a sensory panel consisting of 8 trained members, as described in [32]. The sensory panelists (4 males; 4 females; age range 35 to 52) were recruited based on their previous experience in the sensory evaluation of cereal-based foods among technicians and researchers of the laboratory of the Food Science and Technology unit of the Department of Plant, Soil, and Food Sciences of the University of Bari, Italy. All panel members had neither food allergies nor intolerances and were regular consumers of bread, baked goods, and chickpeas. Pre-test sessions were carried out: (i) to define the list of descriptors to be evaluated in the samples object of the study; (ii) to define the intensity range of each descriptor; (iii) to fix the scale anchors of each descriptor; (iv) to verify reliability, consistency, and discriminating ability of panelists when testing bread, focaccia, and pizza crust. The study protocol followed the ethical guidelines of the laboratory. Panelists were given information about study aims, and individually written informed consent was obtained from each participant. All tested samples were food-grade. A total number of 11 sensory descriptors of appearance, smell, texture, and taste were considered. Seven of them, i.e. external color, chickpea odor, crumb elasticity, crumb consistency, crumb moisture, saltiness, and sweetness were evaluated for all three products; namely bread, focaccia, and pizza crust. Another two descriptors, inner color and crumb porosity, were evaluated only in bread and focaccia due to difficulties in separating the crumb from the surface related to the reduced thickness of pizza crust. Greasiness was evaluated only in focaccia, which contained more oil than the other products, whereas the presence of surface bubbles was evaluated only in pizza crust, where it represents a typical feature. The descriptors were rated on an anchored line scale that provided a 0–9 score range (0 = minimum and 9 = maximum intensity). The analyses were carried out in triplicate.

### 2.11. Statistical Analysis

Statistical analysis was carried out using XLSTAT software (Addinsoft SARL, New York, NY, USA). Significant differences were determined at *p* < 0.05 by one-way analysis of variance (ANOVA) followed by Tukey’s HSD test.

## 3. Results and Discussion

### 3.1. Nutritional Characteristics of the Starting Flours

*Apulian black* chickpea wholemeal flour showed a significantly higher (*p* < 0.05) content of proteins, lipids, and fibre than re-milled semolina (Table 2). The protein content of chickpea flour was slightly higher than those observed in previous works [4,5], whereas the value observed in re-milled semolina agreed with previous quality surveys [22,33]. The fibre content of chickpea flour was within the range observed in other studies [4,5].

The two types of flour exhibited a similar content of phenolic compounds. However, black chickpea flour was characterized by a significantly (*p* < 0.05) higher content of anthocyanins and carotenoids than re-milled semolina. Consequently, the antioxidant activity was also stronger.

These positive features of *Apulian black* chickpea flour confirmed the results of previous studies [4,5].

### 3.2. Rheological and Fermentative Characteristics of Flours and Composite Meals

Composite meals were prepared by mixing *Apulian black* chickpea wholemeal flour with re-milled semolina at 10:90, 20:80, 30:70, and 40:60. The rheological properties of the obtained blends were then evaluated by farinograph, alveograph, and rheofermentograph, in order to assess the suitability to the production of fermented bakery products.

The farinograph parameters measured in pure re-milled semolina agreed with previous works [22]. The addition of black chickpea flour significantly (*p* < 0.05) influenced all farinograph parameters (Table 3). In particular, water absorption, dough development time, and loss of consistency progressively increased, whereas the dough stability decreased. The increase of water absorption was due to the presence of fibre in chickpea wholemeal flour, able to absorb high amounts of water. The dilution of gluten, chickpea flour being gluten free, and the presence of fibre able to interfer with a gluten network, were responsible for the decrease of dough stability, the increase of time needed to develop gluten, and the increase of consistency loss. These results highlight that the addition of chickpea flour worsened the bread-making attitude of wheat flour, with a more evident effect at higher doses. Similar negative effects on farinograph were reported when re-milled semolina was added of other fibre-rich ingredients, such as powdered almond skins [34], or yellow pea wholemeal flour [20]. Moreover, the same effects have been reported for blends of chickpea wholemeal flour with bread wheat flour [14,15] or with semolina for pasta-making [10], evidencing that the bread-making attitude of any gluten-containing flour is depressed by the addition of chickpea flour.

As for the alveograph strength (W), it progressively decreased by the addition of increasing amounts of chickpea flour (Table 3). Moreover, that result was imputable to the increasing content of fibre, contributed by chickpea wholemeal flour, and to gluten’s dilution. The values of the alveograph tenacity/extensibility ratio (P/L) ratio, instead, remained almost constant after the addition of chickpea flour. The value of P/L observed for pure re-milled semolina was in the range observed in previous works, whereas W was particularly high, indicating a very good quality level [22].

The fermentative attitude of flour blends was evaluated by measuring the falling number, related to the amylase activity, and by performing the rheofermentograph analysis. The values of falling number showed a significant difference (*p* < 0.05) only when comparing pure re-milled semolina (with the highest amylase activity) to the composite flour containing the highest chickpea amount (with the lowest amylase activity) (Table 4). The decrease of amylase activity with the addition of 40/100 g of chickpea flour reflected the presence of α-amylase inhibitors, reported in chickpeas and in several other pulses [35]. These inhibitors are slightly more active in *desi* than in *kabuli* chickpea cultivars [36], but can be inactivated by heat treatment [35].

Rheofermentograph data of pure re-milled semolina agreed with other works [37]. Increasing amounts of chickpea flour significantly reduced (*p* < 0.05) the amount of gas produced (*V_T_*) during the rheofermentograph analysis, in agreement with the decrease of amylase activity. Moreover, a greater amount of gas was lost by the dough (*V_L_*) when composite meals were analyzed. Consequently, a progressively lower quantity of gas was retained (*V_R_*) as the addition of chickpea flour increased, reflecting in a significantly lower coefficient of gas retention. Besides, the loss of gas appeared sooner (*T_x_*) in chickpea-added dough than in case of pure re-milled semolina.

The volumetric increase of leavened bakery products depends on both the amount of CO_2_ developed and the rheological properties of dough, in terms of quality and strength of the gluten network, which allows one to effectively retain the gas developed during fermentation. Therefore, these results were due to the diminished fermentative attitude of chickpea-added dough, as shown by the lower amylase activity, and to its weaker gluten network, evidenced by the alveograph and farinograph analyses.

As for the dough development curve, its maximum height (*Hm*), as well as the height after 3 h (*h*), decreased significantly with the increase of chickpea wholemeal flour added, indicating a lower inflation of the dough.

Overall, the bread-making attitude of re-milled semolina worsened by increasing the level of enrichment with black chickpea wholemeal flour. However, studies showed that consumer behavior is changing, driven by awareness of the relationship between nutrition and health. Consequently, whole and fibre-enriched bakery products are accepted better than in the past, despite the lower quality characteristics, particularly if information on the high fibre content is shown on the label [38]. Therefore, the search for the best rheological parameters should no longer be the exclusive criterion for selecting the optimal level of enrichment.

By calculating the theoretical fibre content of the end-products, it was found that the addition of 40/100 g black chickpea wholemeal flour would make it possible to claim the “fibre source” on the label, which requires a content of at least 3 g of fibre per 100 g of product [39]. Lower levels of addition would not reach the conditions for the inclusion of this statement in the label. Considering that information about the presence of fibre influences the acceptance of the modern consumer, leading to a positive perception of unconventional bakery products, the latter were prepared using composite flours of durum wheat re-milled semolina added with 40/100 g of black chickpea wholemeal flour.

### 3.3. Nutritional and Qualitative Properties of the Bakery Products

Table 5 reports the nutritional features of the experimental bakery products (analytically determined data, not calculated). Bread, focaccia, and pizza crust enriched with black chickpea flour showed significantly (*p* < 0.05) higher contents of proteins, lipids, ash, and fibre than conventional products, whereas carbohydrates were significantly lower (Table 5). The slight increase in lipids was coupled to higher fibre content; therefore, the energy value of the enriched products was similar to that of the conventional products.

The addition of black chickpea flour determined a significant increase (*p* < 0.05) of the content of anthocyanins and phenolic compounds in all three baked goods considered (Table 6). However, a marked reduction was observed with respect to the starting flour (reported in Table 2). The decrease of bioactives during thermal processing of *Apulian black* chickpeas was also reported during the preparation of canned sterilized puré [9]. The antioxidant activity, particularly when measured with the ABTS method, was significantly higher in chickpea-enriched bakery products than in durum wheat ones, reflecting the contents of bioactive compounds, specifically anthocyanins and carotenoids.

The variations in the dimensional parameters of bakery products induced by cooking, and their weight loss, are shown in Table 7. Usually, baking involves an increase in volume determined by the thermal expansion of the gases (air trapped during mixing and kneading, water vapor evaporated by the dough, and carbon dioxide originated by leavening). An increase in thickness, indeed, was observed, but without significant differences between conventional durum wheat and chickpea-enriched products. On the other hand, the diameter of circular products, i.e. focaccia and pizza crust, as well as length and width of bread, which had a rectangular shape, decreased with baking. The decrease was significantly more marked (*p* < 0.05) in chickpea-enriched products than in conventional durum wheat products. In bread, the conventional product had an opposite behavior showing an increase of width and maintaining its length almost constant.

Overall, these results agreed with the predictive analyses on dough rheology which, in turn, were due to the interference by fibre and the dilution of gluten. The high alveograph P/L ratio observed in the chickpea-added dough, in particolar, explains the limited expansion during both leavening and the first phases of baking (oven-spring). Thickness was less affected by this limitation because baking essentially induces an upward push [40].

Weight loss though, was not significantly influenced.

Table 8 shows the colorimetric indices of the external and internal surface of the various bakery products prepared. Pizza was inspected only externally, due to its limited thickness (0.7–1.0 cm). The addition of black chickpea flour caused a significant (*p* < 0.05) decrease of *b** and an increase of brown index (100 – *L**) of all the products, which appeared grayish.

Statistically significant differences of *a** were observed only in the internal part of bread and focaccia, being higher in the chickpea-added products. All products from pure re-milled semolina, instead, were bright yellow, with values of *b** almost double those of chickpea-added products. Therefore, together with the reduced expansion degree of products during baking, color alteration is another negative effect of adding black chickpea wholemeal flour, which would require an adequate explanation to the final consumer to highlight the nutritional reasons behind it, in order not to appear too unconventional or even unpleasant (Figure 1).

Table 9 shows the textural parameters of experimental bread and focaccia. Again, pizza crust was not analyzed due to its very limited thickness (0.7–1.0 cm). The chickpea-added bread and focaccia were significantly harder (*p* < 0.05) and more chewy than their counterparts made only of re-milled semolina. The springiness was very similar in all products, whereas the cohesiveness of chickpea-added bread was lower than in conventional durum wheat bread. These results, in agreement with the reduced expansion in volume, can be explained by a worse gluten formation, as already indicated by the alveograph and farinograph parameters of the starting meals, due to the richness in fibre and absence of gluten in chickpea flour.

As for the sensory profile (Table 10), all the chickpea-enriched products were significantly more consistent (*p* < 0.05) than the corresponding conventional products. The addition of chickpea flour determined the significant emergence of an odorous note of chickpeas, a darker color (both external and internal, in agreement with colorimetric data), and an increase in moisture. Saltiness and sweetness, on the other hand, were similar in all the products, as well as crumb porosity.

Greasiness, typical of focaccia but unpleasant if eccessive, was significantly more evident in the conventional focaccia than in the chickpea-added one. The inclusion of oil in the formulation, more abundant in focaccia than in pizza crust and bread, probably influenced the consistency of the focaccia as well, which was slightly softer than bread, in agreement with the results of textural analysis. Other studies reported the positive effect of oil on the consistencies and volumes of bakery products [41].

As for the pizza crust, the presence of surface bubbles was significantly greater in the conventional product than in the chickpea-added one, whose dough was less extensible and less able to retain gas, as shown by rheological analyses.

## 4. Conclusions

On the basis of the results obtained during the characterization of the composite meals, it emerged that the addition of the wholemeal flour of *Apulian black* chickpeas to durum wheat re-milled semolina caused a decrease in the bread-making attitude, which, however, was countered by a nutritional improvement in terms of higher contents of fibre and proteins. The enriched end-products showed also higher contents of bioactive compounds and an improved antioxidant activity. The positive features should be adequately communicated to the consumer to compensate the significant negative effects of addition of chickpea flour, such as alterations of consistency and color with respect to analogous baked goods made of pure re-milled semolina.

Among the three products evaluated, the best product for consumer would be bread because of its lower content of lipids, and consequently, lower energy value. However, with enrichment levels of chickpea flour as high as 40/100 g, all three products evaluated were able to help fulfil the recent dietary guidelines, which suggest one to consume at least three legume servings per week. Adding chickpea flour to baked goods, therefore, represents a nutritionally effective strategy and a significant step forward to increase the consumption of legumes.

## Figures and Tables

**Figure 1 foods-08-00504-f001:**
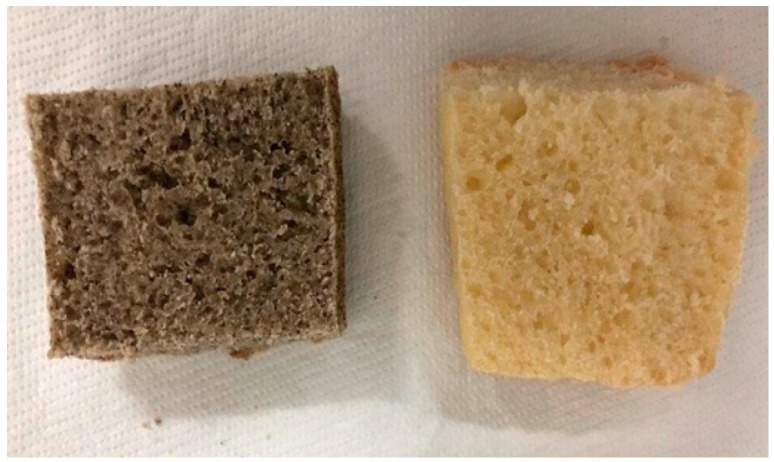
The internal structure of bread prepared by using a composite meal containing 60/100 g of durum wheat re-milled semolina and 40/100 g of *Apulian black* chickpea wholemeal flour (**left**) and bread prepared by using only durum wheat re-milled semolina (**right**).

**Table 1 foods-08-00504-t001:** Formulation of bread, focaccia, and pizza crust, given for 100 g of flour. DW = product prepared by using durum wheat re-milled semolina; BC = product prepared by using a composite meal containing 60/100 g of durum wheat re-milled semolina and 40/100 g of *Apulian black* chickpea wholemeal flour.

Product Type	Ingredient					
	Durum Wheat Re-milled Semolina (g)	*Apulian Black* Chickpea Wholemeal Flour (g)	Water (g)	Extra Virgin Olive Oil (g)	Salt (g)	Yeast (g)
Bread (BC)	60	40	67	-	2	1
Bread (DW)	100	-	62	-	2	1
Focaccia (BC)	60	40	67	20 ^1^	2	1
Focaccia (DW)	100	-	62	20 ^1^	2	1
Pizza crust (BC)	60	40	67	10	2	1
Pizza crust (DW)	100	-	62	10	2	1

^1^ This amount is the sum of 10 g in the dough and 10 g used to oil the pans.

**Table 2 foods-08-00504-t002:** Nutritional characteristics, bioactive compounds, and antioxidant activities of flours used in the production of experimental bread, pizza crust, and focaccia (values are expressed on fresh weight bases).

Parameter	Durum Wheat Re-milled Semolina	*Apulian Black* Chickpea Wholemeal Flour
Moisture (g/100 g)	14.7 ± 0.1 ^a^	9.3 ± 0.2 ^b^
Carbohydrates (g/100 g)	68.5 ± 1.9 ^a^	47.4 ± 1.3 ^b^
Proteins (g/100 g)	12.3 ± 0.1 ^b^	21.4 ± 0.4 ^a^
Lipids (g/100 g)	1.5 ± 0.1 ^b^	4.3 ± 0.3 ^a^
Fibre (g/100 g)	2.1 ± 0.3 ^b^	14.9 ± 1.6 ^a^
Ash (g/100 g)	0.88 ± 0.01 ^b^	2.70 ± 0.01 ^a^
Total anthocyanins (mg/kg cyanidin 3-*O*-glucoside)	n.d. ^1^	69.5 ± 2.6
Total carotenoids (mg/kg β-carotene)	5.75 ± 0.11 ^b^	32.7 ± 2.4 ^a^
Total phenolic compounds (mg/g ferulic acid)	0.97 ± 0.03 ^a^	1.07 ± 0.05 a
Antioxidant activity-ABTS method (µmol Trolox/g)	0.86 ± 0.04 ^b^	1.89 ± 0.07 ^a^
Antioxidant activity-DPPH method (µmol Trolox/g)	2.25 ± 0.29 ^b^	4.04 ± 0.01 ^a^

^1^ n.d. = not detected. Different letters in a row indicate significant differences (*p* < 0.05).

**Table 3 foods-08-00504-t003:** Farinograph and alveograph data of pure durum wheat re-milled semolina and of blends with increasing amounts of *Apulian black* chickpea wholemeal flour.

Parameter	Amount of Black Chickpea Flour (g/100 g)				
	0	10	20	30	40
*Farinograph*					
Water absorption at 500 B.U. ^1^ (g/100 g)	61.8 ± 0.1 ^e^	63.2 ± 0.1 ^d^	64.8 ± 0.1 ^c^	66.3 ± 0.1 ^b^	67.2 ± 0.2 ^a^
Development time (min)	2.0 ± 0.3 ^d^	2.1 ± 0.3 ^d^	4.1 ± 0.3 ^c^	4.9 ± 0.2 ^b^	5.7 ± 0.1 ^a^
Dough stability (min)	8.5 ± 0.5 ^a^	6.6 ± 0.6 ^b^	5.7 ± 0.2 ^c^	3.6 ± 0.2 ^d^	2.6 ± 0.3 ^e^
Loss of consistency at 12 min (B.U.)	49.7 ± 3.2 ^d^	66.3 ± 5.1 ^c^	76.1 ± 3.1 ^b^	83.3 ± 3.8 ^ab^	86.3 ± 1.5 ^a^
*Alveograph*					
W (10^−4^ J)	282 ± 4 ^a^	211 ± 12 ^b^	150 ± 10 ^c^	108 ± 7 ^d^	97 ± 11 ^e^
P/L	2.7 ± 0.1 ^a^	2.6 ± 0.3 ^a^	2.6 ± 0.1 ^a^	2.9 ± 0.3 ^a^	3.0 ± 0.3 ^a^

^1^ B.U. = Brabender units. Different letters in a row indicate significant differences (*p* < 0.05).

**Table 4 foods-08-00504-t004:** The fermentative attitude of pure durum wheat re-milled semolina and of blends with increasing amounts of *Apulian black* chickpea wholemeal flour.

Parameter	Amount of Black Chickpea Flour (g/100 g)				
	0	10	20	30	40
*Amylase activity*					
Falling Number (s)	532 ± 8 ^b^	537 ± 6 ^ab^	541 ± 6 ^ab^	539 ± 3 ^ab^	549 ± 5 ^a^
*Rheofermentograph – Curve of gas production and retention*					
Volume of gas produced (*V_T_*) (mL)	2098 ± 8 ^a^	2077 ± 7 ^b^	2037 ± 3 ^c^	1994 ± 4 ^d^	1972 ± 8 ^d^
Volume of gas retained (*V_R_*) (mL)	1419 ± 11 ^a^	1388 ± 13 ^b^	1325 ± 15 ^c^	1302 ± 21 ^c^	1266 ± 6 ^d^
Volume of gas lost (*V_L_*) (mL)	679 ± 8 ^b^	689 ± 6 ^b^	712 ± 11 ^a^	692 ± 14 ^ab^	706 ± 8 ^a^
Coefficient of gas retention (*V_R_*/*V_T_*) (%)	68 ± 1 ^a^	67 ± 1 ^ab^	65 ± 1 ^bc^	64 ± 1 ^c^	64 ± 1 ^c^
Maximum height of gas production curve (*H’_m_*) (mm)	84 ± 1 ^a^	82 ± 1 ^ab^	81 ± 1 ^b^	74 ± 1 ^c^	73 ± 1 ^c^
Time needed to start losing gas (*T_x_*) (min)	59 ± 2 ^a^	57 ± 4 ^ab^	56 ± 2 ^ab^	53 ± 2 ^b^	48 ± 2 ^c^
*Rheofermentograph – Curve of dough development*					
Maximum dough height (*Hm*) (mm)	49 ± 2 ^a^	44 ± 1 ^b^	41 ± 1 ^c^	35 ± 1 ^d^	23 ± 1 ^e^
Dough height after 3 h (*h*) (mm)	43 ± 2 ^a^	38 ± 1 ^b^	33 ± 1 ^c^	28 ± 1 ^d^	15 ± 1 ^e^

Different letters in a row indicate significant differences (*p* < 0.05).

**Table 5 foods-08-00504-t005:** Nutritional features (values per 100 g, expressed on fresh weight basis) of conventional durum wheat bread, focaccia, and pizza crust and of their black chickpea-enriched versions. DW = product prepared by using durum wheat re-milled semolina; BC = product prepared by using a composite meal containing 60/100 g of durum wheat re-milled semolina and 40/100 g of Apulian black chickpea wholemeal flour.

Parameter	Bread		Focaccia		Pizza Crust	
	DW	BC	DW	BC	DW	BC
Carbohydrates (g)	50.9 ± 2.3 ^a^	44.1 ± 3.1 ^b^	49.1 ± 1.8 ^a^	41.7 ± 2.2 ^b^	52.6 ± 2.6 ^a^	46.2 ± 2.1 ^b^
Proteins (g)	8.5 ± 0.1 ^b^	9.9 ± 0.1 ^a^	7.7 ± 0.1 ^b^	10.3 ± 0.1 ^a^	8.4 ± 0.1 ^b^	10.5 ± 0.1 ^a^
Lipids (g)	0.5 ± 0.1 ^b^	1.3 ± 0.1 ^a^	5.1 ± 0.1 ^b^	5.8 ± 0.2 ^a^	1.9 ± 0.1 ^b^	2.2 ± 0.1 ^a^
Fibre (g)	1.3 ± 0.1 ^b^	5.2 ± 0.3 ^a^	0.9 ± 0.1 ^b^	3.6 ± 0.2 ^a^	1.2 ± 0.1 ^b^	4.6 ± 0.2 ^a^
Ash (g)	1.8 ± 0.1 ^b^	2.1 ± 0.1 ^a^	2.1 ± 0.1 ^b^	2.7 ± 0.1 ^a^	1.8 ± 0.1 ^b^	2.2 ± 0.1 ^a^
Energy value (kJ)	1039 ± 15 ^a^	1008 ± 15 ^a^	1161 ± 19 ^a^	1126 ± 16 ^a^	1117 ± 21 ^a^	1081 ± 16 ^a^

Different letters in a row, for the same bakery product, indicate significant differences (*p* < 0.05).

**Table 6 foods-08-00504-t006:** The bioactive compounds and antioxidant activity of conventional durum wheat bread, focaccia, and pizza crust and of their black, chickpea-enriched versions. DW = product prepared by using durum wheat re-milled semolina; BC = product prepared by using a composite meal containing 60/100 g of durum wheat re-milled semolina and 40/100 g of *Apulian black* chickpea wholemeal flour.

Parameter	Bread		Focaccia		Pizza Crust	
	DW	BC	DW	BC	DW	BC
Total anthocyanins ^1^	n.d. ^5^	7.51 ± 1.47	n.d.	3.49 ± 0.10	n.d.	4.37 ± 0.03
Total carotenoids ^2^	2.69 ± 0.02 ^b^	4.57 ± 0.45 ^a^	2.77 ± 0.01 ^b^	5.21 ± 0.15 ^a^	2.68 ± 0.02 ^b^	5.23 ± 0.16 ^a^
Total phenolic compounds ^3^	0.06 ± 0.01 ^b^	0.09 ± 0.01 ^a^	0.04 ± 0.01 ^b^	0.12 ± 0.01 ^a^	0.07 ± 0.01 ^b^	0.12 ± 0.01 ^a^
Antioxidant activity-ABTS ^4^	1.70 ± 0.05 ^b^	2.13 ± 0.02 ^a^	1.44 ± 0.07 ^b^	1.76 ± 0.03 ^a^	1.25 ± 0.01 ^b^	2.01 ± 0.01 ^a^
Antioxidant activity-DPPH ^4^	0.04 ± 0.03 ^b^	0.08 ± 0.01 ^a^	0.07 ± 0.01 ^b^	0.10 ± 0.01 ^a^	0.08 ± 0.01 ^a^	0.08 ± 0.01 ^a^

^1^ Expressed as mg/kg cyanidin 3-*O*-glucoside (d.m.); ^2^ expressed as mg/kg β-carotene (d.m.); ^3^ expressed as mg/g ferulic acid (d.m.); ^4^ expressed as μmol/g Trolox equivalent (d.m.); ^5^ n.d. = not detected. Different letters in a row, for the same bakery product, indicate significant differences (*p* < 0.05).

**Table 7 foods-08-00504-t007:** Baking-induced variations of the dimensional parameters of conventional durum wheat bread, focaccia, and pizza crust and of their black, chickpea-enriched versions. DW = product prepared by using durum wheat re-milled semolina; BC = product prepared by using a composite meal containing 60/100 g of durum wheat re-milled semolina and 40/100 g of *Apulian black* chickpea wholemeal flour.

Parameter	Bread		Focaccia		Pizza Crust	
	DW	BC	DW	BC	DW	BC
Diameter variation (%)	-	-	−0.7 ± 0.1 ^a^	−3.8 ± 0.1 ^b^	−1.8 ± 0.0 ^a^	−9.8 ± 0.0 ^b^
Length variation (%)	−0.4 ± 0.7 ^a^	−15.1 ± 1.4 ^b^	-	-	-	-
Width variation (%)	7.7 ± 2.1 ^a^	−6.6 ± 3.2 ^b^	-	-	-	-
Thickness variation (%)	84.3 ± 1.6 ^a^	83.1 ± 0.9 ^a^	54.2 ± 1.4 ^b^	59.3 ± 4.1 ^a^	54.2 ± 7.2 ^a^	66.7 ± 5.8 ^a^
Weight loss (%)	10.3 ± 0.3 ^a^	10.2 ± 0.4 ^a^	9.3 ± 0.9 ^a^	9.3 ± 0.6 ^a^	8.9 ± 0.4 ^a^	9.2 ± 0.4 ^a^

Different letters in a row, for the same bakery product, indicate significant differences (*p* < 0.05).

**Table 8 foods-08-00504-t008:** Color parameters of conventional durum wheat bread, focaccia, and pizza crust and of their black, chickpea-enriched versions. DW = product prepared by using durum wheat re-milled semolina; BC = product prepared by using a composite meal containing 60/100 g of durum wheat re-milled semolina and 40/100 g of *Apulian black* chickpea wholemeal flour.

Product Type	Color Parameter		
	*b**	*a**	100 – *L**
*Bread*			
DW ^1^	36.7 ± 3.2 ^a^	6.7 ± 2.2 ^a^	31.7 ± 6.6 ^b^
BC ^1^	21.4 ± 3.7 ^b^	10.3 ± 4.7 ^a^	46.9 ± 3.3 ^a^
DW ^2^	23.9 ± 1.2 ^a^	0.4 ± 0.2 ^b^	19.5 ± 1.9 ^b^
BC ^2^	12.9 ± 0.5 ^b^	1.9 ± 0.1 ^a^	52.7 ± 2.4 ^a^
*Focaccia*			
DW ^1^	38.3 ± 3.1 ^a^	5.7 ± 2.9 ^a^	35.4 ± 7.3 ^b^
BC ^1^	22.5 ± 3.1 ^b^	11.3 ± 6.5 ^a^	51.3 ± 4.9 ^a^
DW ^2^	25.7 ± 0.9 ^a^	0.4 ± 0.2 ^b^	19.5 ± 0.7 ^b^
BC ^2^	16.3 ± 0.7 ^b^	1.7 ± 0.2 ^a^	48.2 ± 1.9 ^a^
*Pizza crust*			
DW ^1^	22.4 ± 2.8 ^a^	2.6 ± 1.3 ^a^	25.1 ± 2.3 ^b^
BC ^1^	12.3 ± 2.3 ^b^	1.2 ± 0.9 ^a^	41.1 ± 8.9 ^a^

^1^ External color. ^2^ Internal color. Different letters in a column for the same bakery product and portion inspected, indicate significant differences (*p* < 0.05).

**Table 9 foods-08-00504-t009:** Textural parameters of conventional durum wheat bread and focaccia and of their black, chickpea-enriched versions. DW = product prepared by using durum wheat re-milled semolina; BC = product prepared by using a composite meal containing 60/100 g of durum wheat re-milled semolina and 40/100 g of *Apulian black* chickpea wholemeal flour.

Parameter	Bread		Focaccia	
	DW	BC	DW	BC
Hardness (N)	7.90 ± 1.54 ^b^	20.80 ± 0.80 ^a^	5.50 ± 0.45 ^b^	14.77 ± 2.01 ^a^
Springiness	0.94 ± 0.01 ^a^	0.92 ± 0.01 ^a^	0.95 ± 0.03 ^a^	0.93 ± 0.02 ^a^
Chewiness (N)	5.59 ± 0.92 ^b^	12.16 ± 1.57 ^a^	3.81 ± 0.35 ^b^	9.90 ± 1.75 ^a^
Cohesiveness	0.74 ± 0.03 ^a^	0.63 ± 0.05 ^b^	0.75 ± 0.05 ^a^	0.72 ± 0.07 ^a^

Different letters in a row indicate significant differences (*p* < 0.05).

**Table 10 foods-08-00504-t010:** The sensory profile of conventional durum wheat bread, focaccia, and pizza crust, and of their black chickpea-enriched versions. DW = product prepared by using durum wheat re-milled semolina; BC = product prepared by using a composite meal containing 60/100 g of durum wheat re-milled semolina and 40/100 g of *Apulian black* chickpea wholemeal flour.

Parameter	Bread		Focaccia		Pizza Crust	
	DW	BC	DW	BC	DW	BC
External color	4.4 ± 0.2 ^b^	7.7 ± 0.2 ^a^	4.5 ± 0.2 ^b^	7.9 ± 0.1 ^a^	3.8 ± 0.2 ^b^	7.2 ± 0.3 ^a^
Inner color	2.9 ± 0.1 ^b^	7.5 ± 0.3 ^a^	3.1 ± 0.5 ^b^	7.1 ± 0.6 ^a^	-	-
Presence of surface bubbles	-	-	-	-	2.1 ± 0.5 ^a^	0.3 ± 0.1 ^b^
Chickpea odor	0.1 ± 0.1 ^b^	4.7 ± 0.6 ^a^	0.2 ± 0.1 ^b^	5.6 ± 0.8 ^a^	0.1 ± 0.1 ^b^	4.8 ± 0.1 ^a^
Greasiness	-	-	5.1 ± 0.2 ^a^	4.1 ± 0.1 ^b^	-	-
Crumb elasticity ^1^	5.7 ± 0.1 ^a^	5.2 ± 0.1 ^b^	6.7 ± 0.4 ^a^	6.3 ± 0.1 ^a^	6.3 ± 1.2 ^a^	5.7 ± 0.9 ^a^
Crumb consistency ^1^	2.9 ± 0.3 ^b^	5.2 ± 0.1 ^a^	1.9 ± 0.2 ^b^	4.6 ± 0.1 ^a^	2.4 ± 0.2 ^b^	4.5 ± 0.3 ^a^
Crumb porosity	4.3 ± 0.3 ^a^	3.9 ± 0.3 ^a^	4.2 ± 0.4 ^a^	4.0 ± 0.1 ^a^	-	-
Crumb moisture ^1^	4.6 ± 0.2 ^b^	5.7 ± 0.2 ^a^	4.3 ± 0.1 ^b^	6.0 ± 0.4 ^a^	4.5 ± 1.1 ^a^	5.7 ± 0.8 ^a^
Saltiness	3.0 ± 0.1 ^a^	2.9 ± 0.4 ^a^	2.9 ± 0.3 ^a^	2.7 ± 0.2 ^a^	2.3 ± 0.2 ^a^	2.1 ± 0.1 ^a^
Sweetness	1.2 ± 0.2 ^a^	1.7 ± 0.3 ^a^	1.3 ± 0.2 ^a^	1.6 ± 0.3 ^a^	1.2 ± 0.3 ^a^	1.6 ± 0.4 ^a^

^1^ Evaluated on the whole product. Different letters in a row, for the same bakery product, indicate significant differences (*p* < 0.05).

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
