# Peer review of "The Effect of the Addition of Apulian black Chickpea Flour on the Nutritional and Qualitative Properties of Durum Wheat-Based Bakery Products"

_foods, 2019, doi:10.3390/foods8100504_

Round 1

Reviewer 1 Report

wheat replacement is not a new idea.  Researchers have been looking at changes in dough functionality with wheat replacers for many years. This paper is another such report.  Like so many of the others the general trend is that increase replacement leads to reduction in quality.

The topic is of moderate interest, but not ground breaking.

was table 5 calculated or determined?

L*a* b* have three axes as continua, what you call a "red index" is a red-green continuum and as all your values are very low it implies they are neither red nor green but close on grey. If your values were close to 100 then using the tern "red index" would make sense.  Similarly the "yellow index" in fact b* is a blue yellow continuum and high values are on the blue side - so it should perhaps be named the "blue index" in the case of your values

How were sensory assessors recruited? Did you seed ethical clearance for the sensory evaluation work? If so, say from where.

Author Response

We thank the Reviewer for careful revision and for helpful suggestions. All observations have been considered, and the text has been modified accordingly. All modifications are highlighted in yellow in the revised version of the manuscript.

Reviewer 1

Point 1. Wheat replacement is not a new idea.  Researchers have been looking at changes in dough functionality with wheat replacers for many years. This paper is another such report.  Like so many of the others the general trend is that increase replacement leads to reduction in quality.

The topic is of moderate interest, but not ground breaking.

Response 1: It is true that wheat replacement is not a new idea, but currently there is great interest in increasing the consumption of legumes and in the scientific literature there are no other articles on the addition of black chickpeas to bread, focaccia and pizza, which could be of great interest for producers because pizza, in particular, is very widespread.

Point 2. Was table 5 calculated or determined?

Response 2: Data in Table 5 were determined. Analytical methods adopted for these data are reported at lines 140-145 of the Materials and Methods section. Preliminary calculations were made for assessing the best percentage of chickpea flour to be added to bakery products so as to state “fiber source” on the label. After those calculations, production trials were made and the obtained products were submitted to nutritional analyses. To avoid confusion, we revised the first sentence presenting Table 5 by adding the specification that data are analytically determined and not calculated (see line 291-292).

Point 3. L*a* b* have three axes as continua, what you call a "red index" is a red-green continuum and as all your values are very low it implies they are neither red nor green but close on grey. If your values were close to 100 then using the tern "red index" would make sense.  Similarly the "yellow index" in fact b* is a blue yellow continuum and high values are on the blue side - so it should perhaps be named the "blue index" in the case of your values

Response 3: We changed the term “red index” and “yellow index” to “a*” and “b*” both in Table 8 and in the text (lines 341-345). Actually, our a* values are close to grey, but our b* values are not on the blue side. The blue side has negative b* values (see, for example, Mortimer, R.J. and Reynolds, J.R., 2005. In situ colorimetric and composite coloration efficiency measurements for electrochromic Prussian blue. Journal of Materials Chemistry, 15: 2226-2233), whereas our values are positive.

Point 4. How were sensory assessors recruited? Did you seed ethical clearance for the sensory evaluation work? If so, say from where.

Response 4: We added some sentences (see lines 178-189) to specify: i) criteria for panel recruitment; ii) steps of selection process; iii) compliance with ethical aspects.

The sensory panelists (4 male, 4 female, age range 35 to 52 y) were recruited based on their previous experience in the sensory evaluation of cereal-based foods among technicians and researchers of the laboratory of the Food Science and Technology unit of the Department of Plant, Soil and Food Sciences of the University of Bari, Italy. All panel members had neither food allergies nor intolerances and were regular consumers of bread, baked goods and chickpeas. Pre-test sessions were carried out: i) to define the list of descriptors to be evaluated in the samples object of the study; ii) to define the intensity range of each descriptor; iii) to fix the scale anchors of each descriptor; iv) to verify reliability, consistency, and discriminating ability of panelists in testing bread, focaccia and pizza crust. Regarding the ethical clearance, we did not seed it, but the study protocol followed the ethical guidelines of the laboratory of the Food Science and Technology unit of the Department of Plant, Soil and Food Sciences of the University of Bari, Italy, inspired by those of the Institute of Food Science and Technology (IFST) (IFST Guidelines for Ethical and Professional Practices for the Sensory Analysis of Foods, https://www.ifst.org/our-resources/ifst-guidelines-ethical-and-professional-practices-sensory-analysis-foods). Panelists were given information about study aims, and an individual written informed consent was obtained from each participant.

The preparation of the experimental bakery products was food-grade being carried out, as stated in the Materials and methods section (lines 105-106), at the bakery laboratory of the Industria Molitoria Mininni-Buéne s.r.l. (Altamura, Italy). This food Company sells flours and bakery products and, besides implementing the HACCP system, its Quality Management System conforms to ISO 9001 and to other certifications in the field of food products such as BRC-Food (British Retail Consortium) and IFS-Food (International Food Standard) which ensure full process standardization and high safety and quality of final products.

As for panelists’expertize, it is documented by a number of articles: Pasqualone, A., Summo, C., Bilancia, M.T. and Caponio, F., 2007. Variations of the sensory profile of durum wheat Altamura PDO (Protected Designation of Origin) bread during staling. Journal of Food Science, 72: S191-S196; Giannone, V., Giarnetti, M., Spina, A., Todaro, A., Pecorino, B., Summo, C., Caponio, F., Paradiso, V.M. and Pasqualone, A., 2018. Physico-chemical properties and sensory profile of durum wheat Dittaino PDO (Protected Designation of Origin) bread and quality of re-milled semolina used for its production. Food Chemistry, 241: 242-249; Pasqualone, A., Caponio, F., Summo, C., Paradiso, V.M., Bottega, G. and Pagani, M.A., 2010. Gluten-free bread making trials from cassava (Manihot esculenta Crantz) flour and sensory evaluation of the final product. International Journal of Food Properties, 13: 562-573; Licciardello, F., Giannone, V., Del Nobile, M.A., Muratore, G., Summo, C., Giarnetti, M., Caponio, F., Paradiso, V.M. and Pasqualone, A., 2017. Shelf life assessment of industrial durum wheat bread as a function of packaging system. Food Chemistry, 224:181-190; Pasqualone, A., Piergiovanni, A.R., Caponio, F., Paradiso, V.M., Summo, C. and Simeone, R., 2011. Evaluation of the technological characteristics and bread-making quality of alternative wheat cereals in comparison with common and durum wheat. Food Science and Technology International, 17: 135-142; Pasqualone, A., Gambacorta, G., Summo, C., Caponio, F., Di Miceli, G., Flagella, Z., Marrese, P.P., Piro, G., Perrotta, C., De Bellis, L. and Lenucci, M.S., 2016. Functional, textural and sensory properties of dry pasta supplemented with lyophilized tomato matrix or with durum wheat bran extracts produced by supercritical carbon dioxide or ultrasound. Food Chemistry, 213: 545-553.

Reviewer 2 Report

This study tests effect of the addition of chickpea flour on physical and nutritional properties of bakery products. The idea is very interesting and original. The introduction provides a sufficient background of the application of chickpea and the effect of chickpea flour addition on bread quality.  The objective of the study is clearly defined.

The experimental apparatus is standard and is appropriate for the study. The methods are well described and provide sufficient information for a capable researcher to reproduce the experiments. The results are clearly explained and presented in an appropriate format.

The conclusions should be better developed. They should involve both positive like and negative effects of addition of chickpea flour. I also suggest to indicate the best product for consumer.

The literature cited is relevant to the study but one item should be replaced (is out of date).

Additional comments for Authors

Page 2, Line 83 – fiber – in British English should be fibre

Page 4, Line 143 “Carbohydrates were calculated by difference” The sentence is not competed.

Page 13, Lines 471-473  -Reference number 28 is out of date

Author Response

We thank the Reviewer for careful revision and for helpful suggestions. All observations have been considered, and the text has been modified accordingly. All modifications are highlighted in yellow in the revised version of the manuscript.

Reviewer 2

Point 1. This study tests effect of the addition of chickpea flour on physical and nutritional properties of bakery products. The idea is very interesting and original. The introduction provides a sufficient background of the application of chickpea and the effect of chickpea flour addition on bread quality.  The objective of the study is clearly defined.

The experimental apparatus is standard and is appropriate for the study. The methods are well described and provide sufficient information for a capable researcher to reproduce the experiments. The results are clearly explained and presented in an appropriate format.

Response 1: We thank the Reviewer for appreciating our effort in making as clear as possible both the experimental plan and the discussion of results coming from this study, involving three different baked goods.

Point 2. The conclusions should be better developed. They should involve both positive like and negative effects of addition of chickpea flour. I also suggest to indicate the best product for consumer.

Response 2: Thank you very much for this helpful suggestion. We added some sentences to highlight also negative effects and to indicate the best product for consumers.

Point 3. The literature cited is relevant to the study but one item should be replaced (is out of date).

Response 3: Thank you very much for your observation, we changed the reference 28 (see below).

Additional comments for Authors

Point 4. Page 2, Line 83 – fiber – in British English should be fibre.

Response 4: Fiber has been replaced with fibre throughout the entire manuscript.

Point 5. Page 4, Line 143 “Carbohydrates were calculated by difference” The sentence is not competed.

Response 5: The sentence has been completed (see lines 143-144).

Point 6. Page 13, Lines 471-473 -Reference number 28 is out of date.

Response 6: Thank you very much for your observation, we changed the reference 28 (Commission Directive 2008/100/EC) with the updated one: Regulation (EC) No 1169/2011, which establishes in Annex XIV the conversion factors to be used for the calculation of the energy content to be declared in food labels (see lines 145-146 and final list of references).